# Using the Very Short Form of the Children’s Behavior Questionnaire for Spanish-Speaking Populations in Low- and Middle-Income Countries: A Psychometric Analysis of Dichotomized Variables

**DOI:** 10.3390/children8020074

**Published:** 2021-01-22

**Authors:** Elsa Lucia Escalante-Barrios, Sonia Mariel Suarez-Enciso, Samuel P. Putnam, Helen Raikes, Sergi Fàbregues

**Affiliations:** 1Department of Education, Universidad del Norte, Km.5 Vía Puerto Colombia, Barranquilla 080001, Colombia; 2Department of Educational Psychology, University of Nebraska-Lincoln, 114 Teacher College Hall, Lincoln, NE 68508, USA; marielsuaren@gmail.com; 3Department of Psychology, Bowdoin College, 255 Maine St, Brunswick, ME 04011, USA; sputnam@bowdoin.edu; 4Department of Child, Youth and Family Studies, University of Nebraska-Lincoln, 205 Louise Pound Hall, Lincoln, NE 68588, USA; hraikes2@unl.edu; 5Department of Psychology and Education, Universitat Oberta de Catalunya, Rambla del Poblenou, 156, 08018 Barcelona, Spain; sfabreguesf@uoc.edu

**Keywords:** Children’s Behavior Questionnaire, temperament, assessment, preschoolers, low- and middle-income countries, Colombia, confirmatory factor analysis

## Abstract

While the psychometric properties of the Spanish version of the Very Short Form of the Children’s Behavior Questionnaire (CBQ-VSF) have been assessed in the US and Europe in samples composed of middle- and high-income parents with high levels of education, no studies have tested the instrument in low-income Spanish-speaking populations living in low- and middle-income countries. To fill this gap, our cross-sectional study assessed the psychometric properties of the Spanish version of the CBQ-VSF version in a sample of 315 low-income and low-educated parents with preschool children living in the Caribbean Region of Colombia. While our findings revealed problems that were similar to those identified in previous assessments of the CBQ-VSF Spanish version, they also showed unique problems related to the sociodemographic characteristics of our sample, containing many individuals with a low income and low educational level. Most of the participants gave extreme responses, resulting in a notable kurtosis and skewness of the data. This article describes how we addressed these problems by dichotomizing the variables into binary categories. Additionally, it demonstrates that merely translating the CBQ-VSF is insufficient to be able to capture many of the underlying latent constructs associated with low-income and low-educated Latino/Hispanic populations.

## 1. Introduction

The definition of temperament has been intensively examined and debated over the years [1,2,3,4]. Rothbart and Bates [5] define temperament as individual differences in reactivity and regulation in the domains of activity, affect and attention that are constitutionally based, early-appearing and relatively stable, and that are influenced over time by heredity and experience. These traits have been assessed using different tools, such as observation, laboratory testing, and questionnaires, still the most-widely used instruments in assessing temperament. The most fine-grained temperament questionnaires developed to date are those constructed by Rothbart and colleagues [6,7,8]. One of these is the Children’s Behavior Questionnaire (CBQ) [9], which is a self-report instrument with a seven-point Likert-type response category, completed by parents about their perception of child temperament in children aged from three to eight years.

The standard CBQ assesses 15 dimensions using 195 items. Bottom-up analysis of its factor structure consistently reveals the following common set of three higher-order dimensions: Extraversion/Surgency (S), Negative Affect (NA), and Effortful Control (EC). The length of the standard CBQ was reduced by Putnam and Rothbart [10] to make it useful in questionnaires conditioned by length limitations. The very short form of the standard CBQ (CBQ-VSF) represents a higher-order three-factor-dimension version of the standard form using only 36 items (12 per factor). The CBQ-VSF was assessed by the developers using multiple datasets that included mostly Caucasian, middle- to upper-SES (socioeconomic Status) samples. This version showed satisfactory internal consistency and criterion validity, and exhibited longitudinal stability and cross-informant agreement, which is consistent with the results of the standard CBQ [11]. It is important to note that the developers extracted the CBQ-VSF factors from data collected with the standard CBQ in order to explore the three-factor structure. This source of data may represent a limitation of this study, because the psychometric characteristics may differ when the analyzed data are collected using a directly administered instrument instead of being extracted from another instrument [12] since factors such as fatigue, item ordering, and neighbor items can affect the participants’ answers [13]. On account of the authors’ explicit focus on the three-factor structure in the development of the CBQ-VSF, Putnam and Rothbart [10] additionally evaluated two orthogonal three-factor nested models (with and without errors allowed to correlate) using Confirmatory Factor Analysis (CFA). The CFA showed a marginal fit of the three-factor model when the authors allowed error terms for items from the same scale to correlate, while the model did not fit when item errors were not allowed to correlate.

Language is a system of symbolic communication shaped by the beliefs, identities and world views of a social group. Each language has its particular metaphors, similes and analogies, and embodies figures of speech filtered through the social consciousness of speakers. According to psychometric literature, cultural and social contexts of linguistic communities determine the linguistic frames their members use to describe everyday physical and social reality and the meaning they attribute to words [14]. Since characteristics such as the SES or the ethnic background of a sample population are likely to affect the psychometric properties of a measure, several authors have examined the performance of the CBQ-VSF in several localities, including the U.S., the Netherlands and Spain. A summary of the characteristics of these studies is shown in Table 1.

In the U.S., in order to examine the factor structure of the CBQ-VSF, a number of researchers have directly administered the original English version in ethnically diverse samples of parents with children who have similar age (three to eight years old) and annual household income (high/middle). Sleddens, et al. [15] used a sample of parents of preschoolers who mostly had high educational attainment (college/university degree). Principal Factor Analysis (PFA) yielded 12 factors with eigenvalues greater than 1.00. The retention of the three-factor structure was supported after forcing the original model; and this modification accounted for 25.1% of the total variance. The variance was explained by each factor as follows: S (9.4%), EC (8.4%) and NA (7.3%). The findings of Sleddens, Kremers, Hughes, Cross, Thijs, De Vries and O’Connor [15] revealed that the structure they examined was similar to the original structure proposed by Putnam and Rothbart [10]. In another U.S. study, Allan, et al. [16] also used the CBQ-VSF English version with a similar sample, except for the educational attainment, which was not reported. In this case, the authors found a very poor model fit in the CFA. Additionally, they performed Exploratory Structural Equation Modeling using oblique geomin rotation. The authors compared models with five factors or fewer, with the constraint that at least three items are needed to represent a construct. The majority of items in the CBQ-VSF loaded on more than one factor, while only NA appeared to be more consistent in maintaining its structure.

In the Netherlands, Sleddens, et al. [17] evaluated the psychometric properties of the Dutch version of the CBQ-VSF (seven-point Likert scale) using data extracted from the standard CBQ. The sample included parents of children aged from six to eight years, who mostly had university degrees. Income and ethnicity were not reported in this study. PFA revealed ten factors, and also showed that only 24% of the variance was explained when the three-factor structure was forced. The authors found that some items had negative loadings, while others unexpectedly loaded on other factors. Despite these differences, the authors concluded that factor loadings were similar to the results reported by Putnam and Rothbart (2006).

In Spain, de la Osa, et al. [18] used a sample of 622 parents with preschoolers. Parents were mainly white (89%) and from high SES families (64%). The authors used CFA and PFA with oblimin oblique rotation to assess the psychometric properties of the CBQ-VSF. Similar to previous studies [10,17], the researchers did not directly use the CBQ-VSF; instead, they extracted the data collected using the CBQ-SF (Short Form of the Children’s Behavior Questionnaire) in Spanish and Catalan. They found 12 factors with the CBQ-VSF. These authors also forced the three-factor solution with a proportion of explained variance of 23%. Item loadings were negative, and they loaded on more than one factor. There were also items loading with values lower than expected. Their findings provided validity evidence of the CBQ-VSF for a predominantly white European Spanish sample with a high SES. However, different results might be found with samples of Spanish-speaking individuals with a low SES and from other ethnic groups, such as Hispanic-American populations.

Samples including Latinos/Hispanics with low income and educational levels might give results that differ from the findings of de la Osa, Granero, Penelo, Domènech and Ezpeleta [18], particularly in the case of extreme response patterns. Some authors have found that extreme response is negatively associated with low income and low educational levels; that is, responses from poor and low educated participants tend to be more extreme than those from richer, better educated participants [19,20]. Furthermore, Marin, et al. [21] argue that Latino/Hispanic individuals also tend to choose more extreme responses in questionnaires as compared with non-Latinos/Hispanics. Those authors state that this tendency may be associated with cultural values of Latino/Hispanic people, particularly collectivism, because this behavior is used to facilitate fluidity and responsiveness in interpersonal interactions that favor the group’s needs. Those authors carried out secondary analyses with four data sets containing responses from Latinos/Hispanics and non-Latino/Hispanic whites. They found that not only were Latinos/Hispanics more likely than non-Latino/Hispanic whites to prefer extreme responses, but also, among the Latinos/Hispanics, those with lower levels of education were more likely to make extreme choices. While the CBQ-VSF using the seven-point Likert scale has been psychometrically tested with low-income Latino/Hispanic Americans in the U.S. [22,23], the Spanish form has not yet been tested with low-income Latino/Hispanic populations living in low- and middle-income countries. To fill this gap, the purpose of this cross-sectional study was to examine the psychometric properties of the Spanish version of the CBQ-VSF in a low-income and low educated sample of Colombian parents with preschool children. Given the large number of families with a low SES in Latin American countries, our findings could lay the foundation for working towards a more fine-grained measurement of temperament in low-income Latino/Hispanic children.

## 2. Materials and Methods

### 2.1. Sample

The sample consisted of 315 primary caregivers of children aged three to five years enrolled in public childcare centers in the Colombian Caribbean region. Fifty-eight per cent of the children were female. The centers are part of the National Early Childhood Education (ECE) program “Cero a Siempre” (From Zero to Always) that serves young children and families living in poverty. In the present study, participants belonged to social strata one and two. In Colombia, the population is classified in six SES strata, hierarchically defined based on the types of dwellings and their urban and rural surroundings. The two lowest strata include individuals who receive public housing subsidies. These two strata are frequently associated with poorly planned urban environments in marginal areas, and the inhabitants are usually among the population with the lowest SES. Most of the caregivers worked mainly part time (58%). Caregivers were mainly mothers (84%). There was a higher percentage of caregivers with only eight years of schooling or less (44%) than with some level of high school (42%).

### 2.2. Measure

The Spanish version of the seven-point Likert scale of the CBQ-VSF, translated by the Research Group in Child Psychology (GIPSE) from the University of Murcia (Spain), was used. Linguistic adaptations for the context were included after back-translating into Spanish (Colombian) and having the text revised by two bilingual experts in health and education. The instrument includes three scales (EC, NA, and S). Parents were asked to report using a seven-point Likert scale ranging from one (extremely untrue) to seven (extremely true). The “not applicable” option was included. Maximum likelihood imputation was used for the item-missing responses, and factor scores were calculated as the mean of all applicable items.

### 2.3. Data Collection

Participants were recruited individually by data collectors at the ECE centers. A total of 405 parents received a letter of invitation, and 78% agreed to participate and signed the consent form. We also used demographic information and health and nutritional data (e.g., BMI, school menu) collected by the centers. Participants received no compensation or incentives. They individually completed the CBQ-VSF in a quiet room at the ECE centers, with professionals available to answer any questions. The data were gathered as part of the first author’s doctoral dissertation. The study procedures were approved by the Institutional Review Board of the University of Nebraska-Lincoln (20150615333EX).

### 2.4. Data Analysis

Internal consistency reliability was evaluated using Cronbach’s alpha and average corrected item-total correlations. CFA was performed to evaluate the three-factor structure and the relationships between the theoretical factors proposed by Putnam and Rothbart [10] for the CBQ-VSF in this particular group of respondents. While previous CFA studies have usually treated the item responses to the seven-point Likert scale of the CBQ-VSF as categorical data [16,26], only Allan, Lonigan and Wilson [16] have explicitly reported treating these types of responses as continuous data. Since the data used for the present study had already been previously treated as categorical by the first author in a study examining the three-factor and single factor models of the CBQ-VSF [24], in the present study we tested two alternative models of data treatment.

Continuous Data Model. The CBQ-VSF was treated as continuous data. Model fit was evaluated using chi-square test (χ^2^). Hu and Bentler [27] guidelines for simple models (i.e., not using cross-factor item loading in the model) were also used when χ^2^ failed to report good fit. The following tests were performed: Comparative Fit Index (CFI), Root Mean Square Error of Approximation (RMSEA), and Standardized Root Mean Square Residual (SRMR). Perfect model-fit was assumed when χ^2^ was non-significant, while it was considered good when CFI ≥ 0.95, SRMR ≤ 0.08 and RMSEA ≤ 0.06 when data were treated as continuous. To calculate this Model, we used Mplus 7.0 with robust maximum likelihood (MLR) estimator.

Categorical Data Model. The respondents in the current study tended to report extreme responses on the seven-point Likert-scale. To overcome this tendency, item responses were dichotomized. Despite its limitations, using dichotomization is justifiable when the distribution of responses is very highly skewed [28]. This is the first time that dichotomization has been used to assess CBQ-VSF. Item categories between one (extremely untrue) and three (slightly untrue) were grouped into one category (untrue), and category responses between five (slightly untrue) and seven (extremely true) were grouped into another category (true), while category response four (neither true nor untrue) was ignored (treated as missing). This adjusted model was then assessed, treating data as ordered categorical during the analysis. In this case, Yu [29] guidelines for Weighted Root Mean Square Residual (WRMR) < 0.90 [30] were used instead of SRMR. An exploratory factor analysis (EFA) was performed when CFA failed to review the number of factors into which the items were distributed. The robust weighted least square (WLSMV) estimator, along with delta parameterization, were used in this model. WLSMV performs well with ordered categorical data [31], small and moderate sample size [32], and shows superior model fit and more precise factor loading when the number of item-response options are low [33].

Data from eight respondents was discarded on account of incomplete questionnaires. Before response 4 was categorized as missing, the item-level missingness varied between 0.65% and 4.2%, and the person-level missingness varied between 0% and 97.2%, with 71% of the participants having no missing responses and 14% having only one missing response. No special treatment was given to missing values on account of the low ratio of missing values per participant (90% of participants have less than three non-responses). In the analysis, we ignored «does not apply» responses since this response was seldom selected by the participants. We did not remove any items from the analyses, because the aim of the study was to assess the original three-factor model structure in Colombia.

## 3. Results

Table 2 shows the descriptive statistics presented by item and scale. The item-level average was relatively high in all cases, ranging between 3.25 and 6.25, while dispersion ranged between 0.94 and 2.19. Items showed positive (15 items) and negative (21 items) kurtosis values, and most items were negatively skewed (29 items). For instance, 57.5% of participants chose the same item category. Cronbach’s alpha values for EC and S were 0.6, which is considered acceptable [34,35], while the values for NA were lower. The corrected item-total correlation varied between −0.18 (Item 20) and 0.41 (Item 31) and item removal did not meaningfully improve internal consistency. Although not reported in the table, Cronbach’s alpha values did not improve after dichotomizing the items. These values remained around the same as those reported in Table 2.

Table 3 shows the results of the CFA for Continuous Data Model and Categorical Data Model.

Continuous Data Model. The factor structure was tested considering data as continuous [16,36]. The three-factor model did not fit the data (χ^2^(519) = 1091.77, *p* = 0.000, CFI = 0.68, RMSEA = 0.06, SRMR = 0.09) even after allowing within-factor error correlations indicated by modification indices. Consequently, each factor structure was independently assessed in individual models. None of the single-factor models fit the data without allowing errors to correlate, as reported in Table 3. After allowing item errors to correlate, goodness of fit indices for EC (χ^2^(46) = 56.54, *p* = 0.14, CFI = 0.97, RMSEA = 0.03, SRMR = 0.04) and NA (χ^2^(49) = 49.71, *p* = 0.44, CFI = 0.99, RMSEA = 0.01, SRMR = 0.04) were good. However, two items (27 and 30) in EC were not statistically significant, while in NA, reversed items were problematic (Item 26 was not significant, and Items 20 and 29 were negatively related to the factor). Additionally, the explained item variance (*R*^2^) ranged between 0.00 and 0.33 in NA and between 0.00 and 0.30 in EC, indicating that more than 66% of the item variances were not accounted for by the factors. The model for S did not fit (χ^2^(40) = 67.01, *p* = 0.000, CFI = 0.94, RMSEA = 0.05, SRMR = 0.06) with four non-significant items and eight negative factor loadings.

Categorical Data Model. The factor structure was tested considering data as categorical. However, since some items included categories with a very small number of cases, data were dichotomized in order to increase the number of cases per category. After this adjustment, the three-factor model did not fit the data (χ^2^(592) = 1080.82, *p* = 0.000, CFI = 0.63, RMSEA = 0.05, WRMR = 1.57) and thus single-factor models were tested individually as before. The EC model fit the data based upon the chi-square test (χ^2^(54) = 68.85, *p* = 0.08), but the alternative fit indices were poor (CFI = 0.82, RMSEA = 0.03, WRMR = 0.92). Two items (24 and 33) did not significantly load on the factor, which were also the ones with the highest level of concentration in original categories five to seven (98% and 95%, respectively). The NA model also did not fit the data (χ^2^(54) = 90.90, *p* = 0.000, CFI = 0.90, RMSEA = 0.05, WRMR = 1.01), and had all three reversed items (20, 26, and 29) with negative factor loadings. Likewise, there was no model-data fit for S (χ^2^(55) = 503.97, *p* = 0.000, CFI = 0.36, RMSEA = 0.16, WRMR = 2.69). Finally, for the purpose of completeness, an EFA was carried out to show the number of orthogonal factors using the scree-plot approach [37]. The number of resulting factors was six when this approach was used, treating the data as either continuous or categorical.

## 4. Discussion

### 4.1. Summary of Results

The primary aim of this cross-sectional study was to examine the psychometric properties of the Spanish version of the CBQ-VSF in a sample of parents living in poverty in the Colombian Caribbean Region, with a low level of educational attainment and preschool children. While the CBQ-VSF has been previously administered to low-income families in North America [22,25,38], to our knowledge, this is the first time that the CBQ-VSF has been psychometrically tested with a sample of Spanish-speaking caregivers living in poverty.

Classical test theory. Internal consistency reliability was evaluated by calculating both Cronbach’s alpha and average corrected item-total correlation. Corrected item-total correlations with values of 0.30 or larger were considered good, while those with values between 0.15 and 0.30 were considered minimally acceptable [15,39]. In this study, most of the corrected item-total correlations were above 0.15, suggesting adequate or sufficient consistency of items. On the other hand, some reversed NA items such as Items 20, 26 and 29 were considered unreliable, suggesting a lack of homogeneity of the items within a scale [39]. Similar results were reported for Item 7 and 13 from S. Cronbach’s alpha values of EC and S were acceptable [34,35]. Previous psychometric studies of the CBQ-VSF with different or similar samples and administration procedures showed higher Cronbach’s alphas for both scales, except for Allan, Lonigan and Wilson [16], who reported similar Cronbach’s alpha for EC. On the other hand, Cronbach’s alpha for NA showed a lower value that is also consistent with the coefficient reported by Allan, Lonigan and Wilson [16]. This low alpha value may be due to a poor correlation between items, suggesting that the structure of existing items and the one-dimensionality of the test would need to be revised. It is important to highlight that item removal did not contribute to meaningful improvement of internal consistency. Given that the majority of participants chose the same item categories, many items were skewed (29 items). When the seven-point Likert scale was transformed into dichotomous variables, Cronbach’s alpha did not improve after dichotomizing the items and remained around the same values.

Confirmatory Factor Analysis. Although seven-point Likert scales are framed within categorical answers, in some empirical studies these scales are treated as continuous, showing tolerable results [36]. Some studies focused on the CBQ-VSF, such as Allan, Lonigan and Wilson [16], have assessed the three-factor model, treating the data as both categorical and continuous. These authors found that the model provided a poor fit to the continuous data, while it also did not show a good fit for categorical data. A previous study of the dataset used in the current study [24] revealed that the three-factor model and the single-factor models of the CBQ-VSF did not show a good model fit when the data was treated as categorical, even after allowing within-factor error correlations indicated by modification indices. In the present study, when data were treated as continuous, the three-factor model did not initially show a good model fit. After forcing the original model, the goodness of fit indices for EC and NA were good. Consequently, these two broad temperamental traits can be captured with the Spanish version of the CBQ-VSF among populations having characteristics similar to those of our sample [40]. Thus, our findings support using the Spanish version of the CBQ-VSF for continuous data to evaluate NA and EC, which are considered, respectively, a risk factor for behavioral problems, and an indicator of voluntary behavioral control [41]. Moreover, studies carried out using a sample of Colombian low-income caregivers have shown a sufficiently good model fit for NA, both when data were treated as categorical (as in Escalante-Barrios [24]), and also when they were treated as continuous (as in the present study). Researchers interested in examining the individual differences of this dimension of temperament among Spanish-speaking populations in low- and middle-income countries might be able to benefit from our findings.

Despite the acceptable results of the Continuous Data Model, the CFA revealed that Item 27 (“Sometimes becomes absorbed in picture book and looks at it for a long time”) and Item 30 (“Approaches places s/he has been told dangerous slowly and cautiously”) in EC, and Item 26 (“Is not afraid of the dark”) in NA were not statistically significant. These findings were also reported by de la Osa, Granero, Penelo, Domènech and Ezpeleta [18], who similarly noted a negative loading for Item 20 (“Hardly ever complains when ill with a cold”) in NA. Sleddens et al. (2012) also found problems with Item 30. In terms of loading, Item 29 (“Is not very upset at minor cuts or bruises”) was negatively related to NA in this study. Even so, the three reversed items of this scale (20, 26 and 29) were problematic, and this effect may be associated with the difficulty Romance language speakers (i.e., Spanish, Italian, French) have in interpreting the double negation in questionnaires [42,43]. Recently, Clark, et al. [44] used Item Response Theory (IRT) methods to identify unneeded items from the original form of the CBQ (195 items), and these can be removed without significantly limiting the reliability, precision and content coverage of the measure. Those authors reported a total of 77 items considered redundant. In our study, we identified as problematic items, Items 20, 26 and 27 of the CBQ-VSF, similarly identified by Clark, et al. [44] as weaker items in the original form of the CBQ. Further analysis using IRT methods might be useful in confirming whether the problematic items in the CBQ-VSF can be removed without detriment to the psychometric functioning of the measure used to assess the higher order temperament factors while at the same time improving the balance between length and quality for the sake of greater parsimony [44]. IRT might also be useful in examining the items of the temperamental trait Surgency, which did not fit either the Continuous Data Model or the Categorical Data Model, for the purpose of determining psychometric functioning at the item and scale level and improving the performance of this scale.

### 4.2. Implications of the Study

Despite the sociodemographic differences between our sample and that used by de la Osa, Granero, Penelo, Domènech and Ezpeleta [18], findings of both studies revealed similar problems related to factor loadings and the three-factor structure of the CBQ-VSF Spanish form, suggesting that linguistic, semantic and grammatical issues might explain to some extent the psychometric properties of this form. However, our study pointed out some unique problems related to the characteristics of our sample (low education and income), which caused several items to have categories with a very small number of cases, owing to the respondents’ tendency to choose extreme responses, and this effect increased the skewness and kurtosis. Consistently, Culpepper and Zimmerman [45] reported that one of the difficulties of surveying low educated Latino/Hispanic samples is their tendency to report extreme responses in seven-point Likert scales. Therefore, in this study the values were dichotomized because the majority of the participants chose the same item categories (29 items were negatively skewed). While the Categorical data model with dichotomized responses did not fit in this study, findings revealed that the number of indicators per factor were increased and the CFA was improved. Dichotomizing variables may help address some limitations found in the literature related to violations of normality of the CBQ–VSF, including skewness and kurtosis, failure of the seven-point Likert response format to work well for all items, and fatigue of respondents when facing seven options to choose from [15,26]. In order to make the items easier to understand for Spanish-speaking, low-income and low education respondents, future studies should disaggregate the items into simpler binary categories that measure the underlying latent constructs of temperament scales in larger samples with similar characteristics [46].

Since some items in our study included categories with very few cases, we dichotomized the data to increase the number of cases per category. Researchers might be able to develop a new Spanish version of the CBQ-VSF that uses binary response categories or five-point responses since some categories in the seven-point response scale are often not selected by participants [44]. While the methodological literature suggests negative consequences of dichotomization -specifically, loss of information about individual differences and effect size, and a decrease in measurement reliability- using dichotomization is justified in two particular situations [28]. One of these when taxometric analyses strongly support the existence of two taxons within the sample. The second instance occurs when the distribution of the variable is very highly skewed. This second example applies to our present study, in line with other studies that have examined the tendency of Latino/Hispanic individuals to choose extreme responses in questionnaires. Marin, et al. [21] suggested that this tendency may be associated with cultural values of Latinos/Hispanics, especially those related to collectivism. In collectivistic societies, individuals tend to respond in favor of the group’s needs, avoiding moderate personal positions and firmly supporting accepted values in order to be responsive in their interpersonal interactions. Disaggregating items into simpler binary categories may help overcome limitations inherent in dichotomizing continuous variables [47], and making the items easier to respond to in the case of Spanish-speaking populations in low- and middle-income countries. This analysis could well be applied to other cultural groups with high collectivistic values.

### 4.3. Limitations of the Study and Future Directions

It is difficult to generalize from our results on account of the homogeneity of the sample in terms of educational attainment, income, cultural background and geographical location. Accordingly, further evidence is needed to definitively validate the Spanish form of the CBQ-VSF. For instance, future research could include adjustments of the form related to simplification of vocabulary, double negation, culturally responsive wording, and the use of seven-point Likert scales. Our study has shown that merely translating and adapting the CBQ-VSF is insufficient to capture the underlying latent constructs in all different linguistic and cultural groups, particularly in relation to the Latino/Hispanic population. Therefore, further psychometric analysis is needed in order to create culturally relevant and accurate versions of the measure.

## Figures and Tables

**Table 1 children-08-00074-t001:** Summary of the characteristics of studies assessing the psychometric properties of the CBQ-VSF.

Author/s, Year, Country, Language	Type of Participants	Children’s Age (Mean)	Income of the Participants	Parents’ Education	Ethnic Group	Type of Analysis	Alpha Values for Each Scale (VSF)
Sleddens, Kremers, Candel, De Vries and Thijs [17], 2011, Netherlands, Dutch	Parents	84.2 months (*SD* = 7.1)	Not reported	A high percentage of mothers (33.5%) and fathers (41.0%) were in possession of a college or university degree	White (100%)	Exploratory factor analysis, coefficient of congruence (to examine cross-cultural differences)	NA (0.72), S (0.76), EC (0.72)
Sleddens, Kremers, Hughes, Cross, Thijs, De Vries and O’Connor [15], 2012, USA, English	Caregivers (parents or guardians)	46.8 months (*SD* = 9.6)	46.0% of the participants had an annual household income above $60,000	30.4% were college graduates, and 28.7% had post-graduate studies	White or European American (39.2%), Black or African American (23.6%), Latino/Hispanic (25.3%), Other (10.5%), No response (1.3%)	Classical test theory, exploratory factor analysis, item response modeling	NA (0.74), S (0.78), EC (0.69)
Allan, Lonigan and Wilson [16] 2013, USA, English	Parents and teachers	53.4 months (*SD* = 7.9)	Average annual family income was $63,329 (*SD* = $40,436)	Not reported	White or European American (59.6%), Black or African American (28.5%), Asian (2.2%), Latino/Hispanic (1.4%), Other or biracial (8.3%)	Confirmatory factor analysis, exploratory factor analysis, convergent and discriminant correlations	Teachers: NA (0.75), S (0.82), EC (0.87); Parents: NA (0.58), S (0.82), EC (0.68)
de la Osa, Granero, Penelo, Domènech and Ezpeleta [18], 2013, Spain, Spanish	Parents and teachers	35.6 months (*SD* = 1.92)	33% of the children’s’ families had a high income, and 31.4% a mean-high income	Not reported	White (88.9%), Latino/Hispanic (7.9%), African (0.3%), Asian (1.0%), Other (1.9%)	Exploratory structural equation modeling, confirmatory factor analysis, principal axis factor analysis	NA (0.77), S (0.65), EC (0.66)
Escalante-Barrios [24], 2016, Colombia, Spanish	Parents and teachers	Between 36 and 60 months	Low income	Colombia: 68.9% of the caregivers had less than a high school diploma, and 14% of the teachers had less than a bachelor’s degree; United States: 35.3% of the caregivers had less than a high school diploma, and 100% of the teachers had a bachelor’s degree	Latino/Hispanic (100%)	Confirmatory factor analysis, regression analysis, path analysis	Colombia: NA (0.5); United States: EC (0.72)
Liu, et al. [25], 2020 USA, English	Teachers	55.5 months (*SD* = 3.58)	96% of the preschoolers had parents who were economically disadvantaged	Not reported	Latino/Hispanic (98%)	Confirmatory factor analysis	NA (0.80), S (0.82), EC (0.80)

**Table 2 children-08-00074-t002:** Descriptive statistics and classical test theory indices of CBQ-VSF, raw data (*N* = 315).

Items	Mean	*SD*	Kurt	Skew	Miss	MRF	ITC	CITC	CAID
Effortful control → Cronbach’s alpha = 0.6									
CBQ3	6.03	1.14	3.94	−1.72	1.95	41.33	0.51	0.34	0.55
CBQ6	5.20	1.70	−0.07	−0.92	3.91	28.88	0.53	0.34	0.54
CBQ9	6.12	1.19	4.73	−1.98	0.65	49.34	0.49	0.33	0.55
CBQ12	6.11	1.34	4.68	−2.16	1.63	51.50	0.57	0.34	0.54
CBQ15	5.62	1.60	1.56	−1.43	3.58	36.61	0.49	0.28	0.55
CBQ18	5.59	1.43	0.83	−1.11	2.28	31.77	0.51	0.29	0.55
CBQ21	6.35	1.00	6.92	−2.32	1.63	57.48	0.43	0.32	0.55
CBQ24	6.29	0.94	6.52	−2.03	1.95	50.68	0.42	0.29	0.56
CBQ27	4.90	1.84	−0.34	−0.82	2.28	27.61	0.34	0.08	0.61
CBQ30	4.50	2.05	−1.16	−0.48	2.93	27.80	0.33	0.06	0.62
CBQ33	6.05	1.21	3.53	−1.78	1.63	44.70	0.36	0.26	0.56
CBQ36	5.42	1.83	−0.15	−1.05	1.30	38.21	0.51	0.28	0.55
Negative affect → Cronbach’s alpha = 0.5									
CBQ2	5.15	1.46	1.00	−1.17	3.26	35.03	0.46	0.33	0.43
CBQ5	4.77	1.69	−0.54	−0.67	2.93	31.62	0.43	0.28	0.43
CBQ8	5.45	1.61	0.68	−1.21	0.98	33.00	0.35	0.19	0.46
CBQ11	4.59	2.19	−1.23	−0.51	2.93	24.23	0.48	0.22	0.45
CBQ14	4.53	1.94	−1.12	−0.41	1.95	24.00	0.55	0.36	0.40
CBQ17	4.87	1.67	0.09	−0.88	2.93	27.46	0.39	0.21	0.45
CBQ20R	3.25	1.90	−1.03	0.51	1.95	28.00	0.06	−0.18	0.56
CBQ23	4.18	1.89	−1.17	−0.11	1.30	17.67	0.50	0.30	0.42
CBQ26R	3.69	2.15	−1.40	0.24	1.30	20.53	0.33	0.07	0.50
CBQ29R	3.34	1.82	−0.77	0.63	3.91	30.80	0.19	−0.04	0.52
CBQ32	5.38	1.76	0.14	−1.09	0.98	33.00	0.49	0.34	0.41
CBQ35	4.74	2.03	−1.13	−0.48	1.63	26.58	0.48	0.25	0.44
Surgency → Cronbach’s alpha = 0.6									
CBQ1	4.49	1.89	−0.94	−0.46	4.23	22.73	0.43	0.28	0.59
CBQ4	5.03	1.89	−0.45	−0.88	2.61	31.29	0.38	0.24	0.60
CBQ7	5.28	1.59	0.47	−1.05	4.23	30.25	0.29	0.11	0.61
CBQ10	5.70	1.47	1.12	−1.29	0.98	37.29	0.43	0.24	0.60
CBQ13R	3.91	2.00	−1.27	0.09	1.95	19.60	0.33	0.11	0.62
CBQ16	4.32	2.06	−1.28	−0.25	3.58	21.23	0.42	0.25	0.59
CBQ19R	4.31	1.92	−1.37	−0.06	2.93	21.02	0.51	0.36	0.57
CBQ22R	3.55	1.94	−1.14	0.37	3.26	24.41	0.40	0.27	0.59
CBQ25	6.00	1.43	3.45	−1.92	0.65	48.85	0.42	0.24	0.60
CBQ28	4.26	2.06	−1.29	−0.28	2.93	21.11	0.44	0.27	0.59
CBQ31R	4.14	2.06	−1.47	0.01	3.58	21.50	0.56	0.41	0.56
CBQ34R	4.19	2.10	−1.44	0.02	1.95	21.48	0.53	0.36	0.57

*SD*: item standard deviation; Skew: skewness; Kurt: Kurtosis; Miss: percentage of missing responses; MRF: highest relative frequency among item categories; ITC: item-total correlation; CITC: corrected item-total correlation; CAID: Cronbach’s alpha if item is deleted.

**Table 3 children-08-00074-t003:** Model-fit statistics for the CBQ-VSF (*N* = 315).

CFA Models	χ^2^	*df*	*p*	CFI	RMSEA	90% CI	*SRMR/WRMR*	# Neg. Loadings	Item *R*^2^ Range
Effortful control → Cronbach’s alpha = 0.6									
Three-factor model									
without error *r*	172.35	54	0.00	0.64	0.09	(0.071−0.099)	0.07	2	0.00−0.32
with error *r*	1091.77	519	0.00	0.68	0.06	(0.055−0.065)	0.09	11	0.00−0.36
Single-factor model for effortful control									
without error *r*	172.35	54	0.00	0.64	0.09	(0.071−0.099)	0.07	2	0.00−0.32
with error *r*	56.54	46	0.14	0.97	0.03	(0.000−0.049)	0.04	2	0.00−0.30
Single-factor model for negative affect									
without error *r*	101.22	54	0.00	0.82	0.05	(0.037−0.069)	0.06	3	0.00−0.39
with error *r*	49.71	49	0.44	0.99	0.01	(0.000−0.038)	0.04	3	0.00−0.33
Single-factor model for surgency									
without error *r*	255.46	54	0.00	0.53	0.11	(0.097−0.124)	0.12	7	0.00−0.56
with error *r*	67.01	40	0.00	0.94	0.05	(0.026−0.066)	0.06	8	0.00−0.55
Categorical Data Model, Data Treated as categorical (dichotomous), WLSMV estimator, no item-error correlation									
Three-factor model	1080.82	592	0.00	0.63	0.05	(0.047−0.057)	1.57	3	0.00−0.65
Effortful control	68.85	54	0.08	0.82	0.03	(0.000−0.049)	0.92	0	0.04−0.57
Negative affect	90.90	54	0.00	0.90	0.05	(0.030−0.064)	1.01	3	0.06−0.59
Surgency	503.97	55	0.00	0.36	0.16	(0.151−0.177)	2.69	0	0.00−0.58

*r*: correlation; *R*^2^: variance; # neg. loadings: number of negative loadings; 90% CI: confidence interval of RMSEA value.

## Data Availability

The data presented in this study are available on request from the corresponding author. The data are not publicly available due to privacy and ethical restrictions.

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
