# Peer review of "Using the Very Short Form of the Children’s Behavior Questionnaire for Spanish-Speaking Populations in Low- and Middle-Income Countries: A Psychometric Analysis of Dichotomized Variables"

_children, 2021, doi:10.3390/children8020074_

Round 1
Reviewer 1 Report
This is an important prospective study looking at the shorter version of the Children’s Behavior Questionnaire and validity in a lower income Spanish-speaking population from a resource limited area of Colombia. They authors make a good argument for testing the validity because of the differences culturally and how people with low resources and lower education levels may answer questions in different Spanish-speaking low and middle income countries. Previous research shows that questionnaire has been normed in Spain and the US Hispanic populations, which as the authors note have more resources than “developing” or LMIC countries. This study was well thought out, carried out, and the manuscript is well written. The data analysis is complex and I do not have the background in analysis to comment on that. I agree with the limitations stated as well as the well thought out adjustments needed for future research. The authors considered many variables that are important when studying a group of people in LMIC countries. Another important point to consider when doing research studies involving questionnaires in other cultures especially those with limited resources is that some of the questions may not be practical for the group being studied. For example one question discussed in this paper involved looking at a picture book. It has been my experience that populations such as the one used in this study have very limited access to books at all. It is also my experience that parents will answer a question, even if they do not have access to the resource suggested in the question, in a way “guessing”. I suggest the authors consider this additional limitation as a possible reason for results that may not match expectations. I have found it is helpful in reviewing each question with local people in any given resource limited community to see if it is culturally and practically valid. This is an important study and as the authors point out, doing questionnaires in one culture does not allow these questionnaires to be easily generalizable, but the more they are studied in resource limited countries, the better understanding we will have for helping understand the needs in these areas. Specific Comments: Title: I would consider changing “developing countries” in the title and throughout the paper to one of the more currently accepted terms such as “low and middle income countries” (LMIC). Abstract: Well written abstract that is concise and easy to read and understand. Introduction: This is a well-organized summary outlining past studies and the importance of this study. The purpose is clearly stated. Methods: The procedure is clearly written and includes enough information to easily replicate. No recommended changes. Results: The tables and Figures are labeled well and easy to follow and include all relevant information. Discussion: The discussion is well organized and written clearly. It includes relevant limitations and ideas for future research. See comments in general comments regarding additional thoughts.Author Response
Please find the responses to the reviewer's comments in the Word document.

Reviewer 2 Report
This manuscript titled, “Using the Very Short Form of the Children’s Behavior Questionnaire for Spanish-speaking Populations in Developing Countries: A Psychometric Analysis of Dichotomized Variables” presents an analysis of the Spanish version of the CBQ- Very Short Form completed by low-income and low-educated Hispanic participants in Colombia. The results of this study suggest that the measure performed quite poorly and the factor structure of the original CBQ-VSF was not upheld. The manuscript is well-written and the methodology is sound, but I am left wondering what to make of the results of this study since the measure performed so poorly in this sample. The discussion explains why these results may have occurred, but it is still unclear what can be done in the future. In particular, it seems as though the best recommendation is to use dichotomized variables with this sample since many participants report the same response category for items. However, do these dichotomized variables still adequately capture variability in the 3 temperament dimensions? What seems to be missing is a more practical discussion of what can and should be done rather than focusing almost exclusively on the statistics. Additional comments are included below by section.
Abstract
-Lines 22-24: “…assessed the psychometric properties of the Spanish of the CBQ-VSF…” needs to be edited.
Introduction
-The introduction is well-written, though due to the statistical nature of the paper as a whole, it is very statistically focused. The psychometrics of previous analyses of the CBQ-VSF in different samples are presented. The introduction would benefit from an integrative, higher order summary of these results. As written, it is difficult to understand the take home messages from previous studies when the very detailed statistics are primarily what is reported.
Materials and Methods
-Lines 137-138: More information about the SES strata (one and two) is needed. How are these strata defined?
-Lines 149-150: What is the % missingness for the sample? How many participants had missing data and what % of the data were missing?
-Lines 193-194: Is the item-level missingness reported here before or after a response of “4” was considered missing?
Results
-Lines 223-225: Did the model fit improve after the problematic items were dropped? Item removal is discussed in the Discussion section, but it should be reported here instead with more detail. For example, were all problematic items dropped? If not, which ones were dropped?
Discussion
-As described above, what is missing from the discussion is a practical discussion of what can and should be done given that this measure performed so poorly. With the recommendations the authors make (i.e. dichotomizing the variables), does this measure still adequately capture variability in child temperament? What is to be made of the results that the original 3-factor structure was not upheld? Are the factors that did emerge conceptually meaningful?
Author Response
Please find the responses to the reviewer's comments in the Word document.

Reviewer 3 Report
I have completed a review of the manuscript, “Using the Very Short Form of the Children’s Behavior Questionnaire for Spanish-speaking Populations in Developing Countries: A Psychometric Analysis of Dichotomized Variables”. The manuscript strengths include a unique sample from outside the US, sufficient justification for the proposed work, and clear psychometric implications of findings. Further, the authors’ writing is strong and this manuscript was enjoyable to read. As such, I believe that this manuscript contributes to the field and see a definite place for it in the literature.
However, I do see some areas for improvement of the paper. Broadly, I believe the authors should (1) reorganize their introduction and expand on the conceptual framework, (2) add in some detail about the procedure, and (3) strengthen the implications section. I give some suggestions on how to address these points below:
Introduction
- 3 line 110: this section is well-written and central to the paper. It should be moved up in the introduction, perhaps as the third paragraph.
- 3 line 117: the research about extreme responding is very interesting. As above, please move up in the introduction and expand. It seems like the authors expected this and it was foundational for their analytic decisions. I would have liked the introduction to set this up more.
- Much of the intro is dedicated to reviewing prior CBQ-VSF factor solutions. Could the authors put this review into a table? The information is important but slows down the flow of the introduction as is. Putting it into a table format could allow readers to gain this information more quickly.
Method
- Was information collected on participant age, child age, or child sex?
- Was the measure back-translated? Please clarify.
- Please include information about participant recruitment, consent/assent, and compensation.
- 4 line 188: move the information about MLR up to when talking about Model 1.
- 4 line 195: were participants more likely to respond to certain questions with “does not apply”?
- It may be helpful to name the models rather than calling them Model 1 and Model 2. This is personal preference, but will help readers remember which is which!
Discussion
- 8 line 313: Please include information about the implications of the items that were problematic. What does this mean for future research?
- Is qualitative work needed to further probe into the authors findings?
Author Response

(The authors gave the same response as above.)
